# Importance of Metalloproteinase 8 (MMP-8) in the Diagnosis of Periodontitis

**DOI:** 10.3390/ijms25052721

**Published:** 2024-02-27

**Authors:** Emilia Anna Zalewska, Renata Ławicka, Piotr Grygorczuk, Magdalena Nowosielska, Aleksandra Kicman, Sławomir Ławicki

**Affiliations:** 1The “Karedent” Dental Clinic, Bukowskiego 1/u3, 15-006 Bialystok, Poland; emaliazet31@gmail.com (E.A.Z.); rlawicka4@gmail.com (R.Ł.); piotr_grygorczuk@wp.pl (P.G.); 2Department of Gerostomatology, Medical University of Białystok, Akademicka 3, 15-267 Bialystok, Poland; magdalena.nowosielska@umb.edu.pl; 3Department of Aesthetic Medicine, Medical University of Białystok, Akademicka 3, 15-267 Bialystok, Poland; olakicman@gmail.com; 4Department of Population Medicine and Lifestyle Diseases Prevention, Medical University of Białystok, Waszyngtona 13a, 15-269 Bialystok, Poland

**Keywords:** MMP-8, periodontitis, metalloproteinases, collagenase 2, neutrophil collagenase

## Abstract

Periodontitis is a complex condition. Left untreated, it leads to tooth loss and the need for prosthetic treatment. The incidence of periodontitis is steadily increasing, so new methods are being sought to aid in the diagnosis of the disease. Among the methods postulated is the determination of concentrations of bioactive compounds which include extracellular matrix metalloproteinases (MMPs). These enzymes are present in various structural elements of the stomatognathic system. The most promising enzyme of this group appears to be metalloproteinase 8 (MMP-8). MMP-8 assays are performed in gingival fluid or saliva, and MMP-8 levels have been shown to be higher in patients with periodontitis compared to healthy subjects and correlated with some clinical parameters of the condition and the severity of the disease. In addition, the preliminary usefulness of this enzyme in evaluating the effectiveness of periodontal treatment and doxycycline therapy has been demonstrated. Determination of the active form of MMP-8 (aMMP-8) in oral rinse fluid using off-the-shelf assays shows the highest potential. Despite reports about aMMP-8 and promising data on the role of MMP-8 in periodontal diagnosis, a clear determination of the usefulness of this enzyme requires further research.

## 1. Introduction

The periodontium is a tissue complex responsible for attaching the teeth to the alveolar process, and also for their protection and nutrition. The periodontium consists of the gingival complex, alveolar bone, periodontal ligaments, and root cementum [1,2,3]. When periodontal elements become chronically inflamed, periodontal disease develops [1,4,5,6,7,8,9,10]. Untreated periodontal disease is the most common cause of tooth loss, which in turn translates into decreased quality of life and decreased self-confidence in patients [4,6,7,8,9]. Periodontal disease is now a rapidly spreading global problem. It is estimated that 20–50% of the world’s population will develop periodontal disease at some point in their lives, and this percentage is steadily increasing [7,9].

The etiology of periodontitis is complex and multifactorial, but the disease has been shown to be initiated by bacteria present in the plaque that forms on the teeth and gums. With inadequate oral hygiene, plaque bacteria cause gingivitis, which then develops into periodontitis [1,4,5,6,7,8,9]. Many overlapping factors are involved in the maintenance and further development of periodontitis, including genetic and environmental determinants, the host immune response to inflammation, and behavioral factors. Various types of biological active compounds are involved in the complex pathomechanism of this disease, possibly including extracellular matrix metalloproteinases (MMPs) [5,7,10,11,12].

Enzymes from this group are also involved in the pathogenesis of many diseases, such as the development of cardiovascular [13], nervous, or respiratory illnesses [14], and the process of carcinogenesis; their increased activity has been associated with the progression of these conditions [15]. Currently, there are attempts to use MMPs as markers of cancers such as breast cancer or cancers of the reproductive organs [16,17,18]. Some studies indicate that these enzymes can be considered as potential markers in the diagnosis of periodontitis and the prognosis of its development [11], and of particular interest in the field of periodontology is metalloproteinase 8 (MMP-8), which belongs to the group of collagenases. The aim of the present study was to gather current knowledge on the importance of MMP-8 in periodontitis and also the usefulness of this enzyme in the diagnosis of this condition. In addition, the study provides a brief introduction on the etiology of periodontitis and the biological role of metalloproteinases and MMP-8.

## 2. Symptoms and Risk Factors Associated with Periodontitis

In the majority of the population, periodontitis is initially painless, which translates into further progression of the disease, and often leads to tooth loss. The reason for the painless course of this disease is currently not clearly explained; some studies, however, indicate that it is due to a complex mechanism between the host organism and the bacteria present in the oral cavity, as well as the inhibition of pain mediator activity [19]. Despite the absence of pain, patients report a number of symptoms including bleeding during tooth brushing, which is one of the first symptoms suggesting the need to visit the dentist [20]. As the disease progresses, redness and swelling of the gingival area and the presence of gingival pockets are observed. Patients report problems with a change in taste and the occurrence of halitosis, i.e., unpleasant mouth odor. Pain usually occurs with abscess formation and loosening of the tooth attachment to the alveolar ridge, resulting from weakened tooth support [5,20,21]. The destruction of ligamentous structures and alveolar bone results in exposure of the tooth root, increasing tooth mobility and ultimately tooth loss [5,20,21].

Periodontitis risk factors can be divided into modifiable and non-modifiable ones. Some of the most important non-modifiable risk factors include age, gender, and genetic predisposition. In general, as a patient ages, not only does the risk of periodontitis increase, but so does severity, which is most likely related to the senile atrophy of periodontal tissues, the length of time the periodontium has been exposed to bacterial plaque, and a patient’s cumulative history of oral diseases [22,23,24,25]. Although women are more prone to periodontitis due to higher levels of estrogen, which modulates the immune system, and due to changes in hormone levels at different stages of life, men are more likely to be diagnosed with this group of conditions. It is estimated that men have a 50% higher incidence of periodontitis. The most likely reason for this phenomenon is poorer oral hygiene care and less frequent dental visits in male patients. In addition, some authors indicate that the higher frequency of periodontal disease in the male population is also related to lifestyle: the amount of drinking and smoking among men has been shown to be higher than among women [25,26,27,28,29,30]. The role of smoking and alcohol consumption in terms of periodontal disease will be described in later sections of this paper.

Also, genetic conditions can affect the occurrence of periodontitis. According to the literature, patients with known genetic syndromes, especially those related to immune disorders, have an increased risk of developing periodontal disease and the course of the disease is more aggressive. Such genetic syndromes include neurofibromatosis type 1, Papillon–Lefèvre syndrome, and Chèdiak–Higashi syndrome [31,32,33]. Genetic polymorphisms are also associated with an increased risk of periodontitis. According to existing knowledge, variants of at least 65 genes are associated with the risk of periodontitis and polymorphisms, including genes encoding IL-1 (interleukin 1), tissue compatibility system (HLA) molecules, and TNF-α (tumor necrosis factor alpha), among others. Tumor necrosis factor alpha (tumor necrosis factor), the receptor for vitamin D, and the promoter for MPO (myeloperoxidase) have been associated with a more aggressive course of this group of diseases [27,33,34,35]. Also, polymorphisms within genes encoding MMPs are associated with an increased risk of developing periodontitis. The most commonly described polymorphisms related to the above issue mainly involve MMP-2, MMP-3, MMP-9, and MMP-12 [36,37]. The role of polymorphisms in MMP-8 genes in association with periodontitis has also been demonstrated; however, this will be presented in later sections of this paper. 

Smoking is one of the most important modifiable risk factors for periodontal disease. Tobacco smoke contains more than 4000 different toxic compounds that negatively affect the function of the cells and tissues that make up the periodontium. Smokers have a higher risk of periodontal disease compared to non-smokers. Additionally, periodontal disease in smokers has a more severe course with a tendency to recur, and patients do not respond as well to treatment [23,25,38]. Since the compounds in tobacco smoke mask the symptoms of periodontitis, especially bleeding during tooth brushing, patients who smoke may overlook the first signs of the developing condition, which further translates into further progression of the disease [23,38]. According to existing knowledge, the initiation and progression of periodontal disease in smokers is a result of several mechanisms such as suppression of the normal immune response within the oral cavity, the inhibition of regenerative processes within the periodontium, and a decrease in perfusion within the gingiva, which translates into a reduction in oxygen and nutrient delivery while impairing the removal of unnecessary metabolites [38,39,40]. Interestingly, marijuana smokers also have an increased risk of periodontal problems; however, it is not known whether periodontal disease in this case develops through similar mechanisms as in tobacco smokers [39,40,41,42].

Right next to smoking, improper or insufficient oral hygiene is the most important and modifiable risk factor for periodontitis [38,43]. Insufficient oral hygiene, understood as improper or irregular tooth brushing, failure to clean interdental spaces, and avoidance of follow-up dental visits, leads to plaque accumulation [44,45]. Dental plaque is a structurally and functionally organized biofilm containing microorganisms that is deposited on the tooth surface [46]. The presence of certain batteries in bacterial plaque is associated with an increased susceptibility, initiation, and progression of periodontitis. Bacterial species have been divided by color into six different complexes. Bacteria from the so-called red complex are considered to be most significant in the pathogenesis of periodontal disease. This group includes bacteria such as *Porphyromonas gingivalis*, *Treponema denticola*, and *Tannerella forsythia*. Other bacteria, from beyond the red complex, that are important in the pathogenesis of periodontal disease include, among others, *Fusobacterium nucleatum*, *Campylobacter rectus*, and *Peptostreptococcus micros* [46,47]. 

A number of systemic diseases can affect the host immune response and contribute to the risk of periodontal disease. These can include type I and type II diabetes [25]. It has been shown that individuals with diabetes have an almost three times higher risk of periodontitis than healthy individuals [23,26]; on the other hand, periodontitis is also associated with a sequela of poorer glycemic control and a higher incidence of diabetic complications such as retinopathy, neuropathy, nephropathy, and the occurrence of cardiovascular complications [48,49]. According to the literature, periodontal treatment in individuals with diabetes is associated with improvements in glycemic control [49]. The most important modifiable and non-modifiable causes of periodontitis are shown in Figure 1.

## 3. Metalloproteinases and Their Role in the Human Body with Special Emphasis on the Stomatognathic Apparatus

MMPs belong to the group of proteolytic enzymes and the activity of these compounds depends on the presence of zinc ions [13,14,50,51]. In the human body, 23 MMPs have been identified [13]. MMPs, depending on their substrate affinity, are classified into one of six groups, as shown in Figure 2.

Enzymes in this group are produced by a number of cells such as smooth muscle cells, leukocytes, platelets, fibroblasts, and endothelial cells [13,14,50,51,52,53,54,55]. Importantly, the expression and/or secretion of particular MMPs is found in cells that build the dental tissues, such as ameloblasts during enamel development and odontoblasts. These enzymes have also been isolated from dentin, predentine, pulp tissue, and dental fluid [56,57,58,59,60,61,62,63].

The overarching function of MMPs is related to the degradation of extracellular matrix elements and also other proteolytic enzymes, protease inhibitors, blood coagulation factors, cytokines, and growth factors [13,14,50,51]. Physiological MMPs mediate cellular phenomena such as cell differentiation and apoptosis, as well as tissue remodeling and wound healing processes. Also, a number of processes related to reproduction, i.e., ovulation, endometrial reconstruction during the menstrual cycle, childbirth, and fetal development (organogenesis, embryogenesis), depend on the activity of MMPs [13,14,50]. MMPs are also involved in all phases of tooth development, mainly by mediating the processes of proliferation and apoptosis, as well as the degradation and mineralization of tooth tissue. Of particular importance in tooth development is metalloproteinase 20 (MMP-20), which is involved in enamel development [59,63,64,65]. Mutations in the gene encoding MMP-20 have been proven to cause disorders in proper enamel development such as amelogenesis imperfecta. The enamel of patients with this disorder is characterized by a soft and rough surface with foci of pigmentation [65,66]. Additionally, such patients are at risk of more rapid dental caries and earlier tooth loss [67]. Furthermore, some studies indicate that MMPs are involved in tooth eruption [65].

Overactive MMPs lead to tissue destruction, weakening of the extracellular matrix, and fibrosis, which translate into the initiation and progression of various types of diseases. The dysregulation of MMPs is associated with the development of cardiovascular, respiratory [13,51,53,68], nervous [14,51,69,70], and excretory diseases [51,71], among others. Also, in the field of dentistry, the adverse effects of MMPs have been shown to be related to their prolonged activity. These enzymes are associated with caries formation, reversible and irreversible pulpitis, periapical tissue inflammation, and periodontal disease [58,63,72,73].

The negative properties associated with MMP dysregulation are particularly evident at all stages of carcinogenesis. At the tumor-initiation stage, MMPs induce DNA damage by degrading enzymes responsible for repairing defects in the chain. MMPs stimulate the proliferation, invasion, and migration of tumor cells, which, while stimulating angiogenesis, contributes to tumor progression. In turn, the proteolytic properties of these enzymes contribute to the formation of metastatic foci [14,51,74,75,76]. Importantly, the expression of particular MMPs is found in different types of odontogenic tumors, such as ameloblastoma, ameloblastic carcinoma, adenomatoid odontogenic tumors, calcifying cystic odontogenic tumors, and odontomas [77,78,79,80,81]. In patients with calcifying cystic odontogenic tumors, overexpression has been associated with the progression of this type of cancer [79]. A brief summary of the role of MMPs in diseases of the stomatognathic system is shown in Figure 3.

## 4. Structure and Functions of Metalloproteinase 8 (MMP-8)

MMP-8 (collagenase 2/neutrophil collagenase) belongs to the collagenase group and the enzyme was first identified in 1986 in human polymorphonuclear leukocytes [12]. The gene encoding MMP-8 is located on chromosome 11q22.3. Interestingly, this gene is in the same cluster as eight other genes encoding MMP-1 and MMP-13, among others [50,82,83]. The structure of the MMP-8 molecule is shown in Figure 3. 

MMP-8 is secreted as an inactive pro-enzyme mainly by neutrophils, which store the pro-enzyme in specialized granules, and also by endothelial cells, chondrocytes, activated macrophages, and myocytes [12,13,50]. Importantly, MMP-8 is also present in human odontoblasts and is the most important collagenase active in dentin. It is also produced by mesenchymal cells of the dental pulp [12]. The release of pro-MMP-8 from specialized granules occurs under the influence of various bioactive mediators such as interleukin 1, interleukin 8, TNF-α, components of the complement system, fibrin breakdown products, and granulocyte–macrophage colony-stimulating factor (GM-CSF) [12,84]. The inhibition of MMP-8 release is mainly regulated by transforming growth factor β (TGF-β) [12]. The direct activation of pro-MMP-8 to MMP-8 occurs through the action of metalloproteinase 3 and 10 (MMP-3, MMP-10) [13]. Importantly, MMP-8 activation also occurs through bacterial proteases, whose source is the microorganisms present in the bacterial plaque [85].

The most important physiological role of MMP-8 is to facilitate the migration of neutrophils from the circulation into tissues, including periodontal tissues. This is completed by degrading the tissues that make up the extracellular matrix. The physiological function of MMP-8 translates into the broad substrate spectrum of MMP-8 [82]; the enzyme has been shown to be capable of degrading collagen types I, II and III, fibronectin, aggrecan, fibrinogen, bradykinin, and a2-macroglobulin [12,13]. MMP-8 is one of the few enzymes capable of degrading cleaving native fibrillar collagens, which facilitates further degradation of the components of this protein by other MMPs, thereby regulating tissue remodeling [86,87]. This enzyme is also crucial in wound healing and the regulation of the inflammatory process [88]. When MMP-8 is over-activated, this enzyme contributes to the progression of many diseases such as rheumatoid arthritis, asthma, and cancer [13,89]. The particular importance of this enzyme has also been shown in the pathogenesis of periodontitis [84].

## 5. Importance of MMP-8 in Periodontitis

According to Kang et al. [90], MMP-8 is the main proteolytic enzyme detected in periodontal tissue that is affected by inflammation, and its source is most likely degranulating neutrophils. Importantly, bacteria associated with the development of periodontitis, such as *Treponema denticola* and *Porphyromonas gingivalis*, through the action of the proteases they produce, increase the activity of this enzyme [91]. This translates into a high potential for this enzyme as a potential diagnostic marker. Most studies have been conducted in saliva or gingival fluid. In addition, the presence of MMP-8 is also found in parotid and sublingual salivary gland fluid. Concentrations of MMP-8 are most often determined by two methods, enzyme-linked immunosorbent assay (ELISA) or immunofluorometric assay (IFMA), with IFMA having shown better accuracy in determining this compound [92,93]. Studies analyze both the activity of the enzyme (aMMP-8) and its total concentration (tMMP-8). Interestingly, pro-MMP-8 is also found in the gingival fluid [94]. When tested using q-PCR (quantitative polymerase chain reaction), mRNA expression for MMP-8 can also be determined [95].

Gingival fluid is the physiological product of serum exudate that is present in the gingival pocket [93]. Gingival fluid mainly contains electrolytes, proteins, and lipoproteins, and when inflammation occurs, its composition is altered so that it can be used as a non-invasive diagnosis of periodontal disease. The collection of gingival fluid is performed using the intracrevicular washing technique (using two injection needles) or the absorption technique using paper strips placed in the gingival crevice [91,93]. It is also important to note that, according to Ramenzoni et al. [96], the highest amounts of MMP-8 come specifically from gingival fluid. According to most studies, patients with periodontitis have higher concentrations and an elevated enzymatic activity of MMP-8 in the gingival fluid compared to healthy individuals [93,94,96,97,98,99,100,101,102,103,104,105,106,107]. 

According to a study by Fatemi et al. [97], patients with periodontitis had 3.6 times higher concentrations of this enzyme than healthy subjects. Importantly, in addition to the elevated levels of MMP-8 in periodontal patients, the levels of this enzyme correlated with some clinical parameters of periodontitis.

MMP-8 concentrations were observed in subjects with more severe forms of periodontitis compared to those with moderate and mild forms of the disease [93,99,101]. However, it should be noted that a single study indicates that the determination of MMP-8 concentrations is not suitable for differentiating aggressive periodontitis from chronic periodontitis [100]. On the other hand, however, a study by Hernández et al. [93] indicates that MMP-8 has shown high diagnostic accuracy in detecting and differentiating severe and mild periodontitis. 

MMP-8 concentrations positively correlate with clinical parameters of periodontitis severity such as CAL (clinical attachment level) and plaque index (PI). Also, the relationship between MMP-8 expression and CAL index shows a strong positive correlation [97]. As reported by Mauramo et al. [99], Ramenzoni et al. [96], Marcaccini et al. [106], and Chen et al. [108], high levels of MMP-8 in gingival fluid are associated with an increased depth of gingival pockets and the appearance of bleeding during probing.

A single study indicates the potential usefulness of MMP-8 in predicting response to periodontal treatment; in smoking patients, high levels of MMP-8 were associated with poorer response to treatment [109]. However, a clear determination of the usefulness of this enzyme in predicting treatment response will require more extensive studies in the future involving patients with different stages and types of periodontitis. 

Some studies also indicate correlations between MMP-8 levels and the presence of microorganisms involved in the pathogenesis of periodontitis. According to a study by Nędzi-Góra et al. [101], high levels of MMP-8 were associated with the presence of *Fusobacterium nucleatum*, a microorganism that is involved in the pathogenesis of periodontitis. On the other hand, Yakob et al. [110] indicate that the presence of *Tannerella forsythia* and *Treponema denticola* suggests higher levels of this enzyme in the gingival fluid. This supports the theory that bacterial proteases may contribute to higher amounts of active MMP-8 (aMMP-8). 

Importantly, periodontal treatments and the use of oral antibiotics affect MMP-8 concentrations in the gingival fluid. After periodontal treatment—laser or surgical treatment—a decrease in the concentration of MMP-8 in the gingival fluid is found (in studies performed 3 or 6 months after the treatment). However, a greater decrease is observed in the case of diode laser treatment, which indicates that the laser treatment is more effective [111]. Also, performing less invasive procedures such as ultrasonic scaling and curettage translated into a decrease in MMP-8 levels [104,106,108,109,112,113]. In the case of scaling and curettage, the greatest decrease in MMP-8 concentrations was seen after 3 months [106,112]. A study by Emingil et al. [114] also indicates that the use of scaling or curettage combined with low-dose doxycycline is associated with a decrease in MMP-8 concentration in the gingival fluid. These results are also confirmed by Choi et al. [115], who also indicate that the use of doxycycline reduces MMP-8 concentration. These studies indicate the potential of MMP-8 in monitoring the efficacy and course of low-dose doxycycline therapy in patients with periodontitis.

Interestingly, the use of anti-inflammatory drugs such as meloxican in combination with periodontal treatments (scaling and root planning, day 3 after meloxicam therapy) was also associated with a decrease in MMP-8 levels in the gingival pockets of patients. This decrease was best seen on day 10 of therapy [116].

As mentioned earlier, a fair amount of MMP-8-related research has been conducted in saliva. Five minutes before the start of the study, each participant rinsed his or her mouth with water to cleanse it of debris. Saliva was collected in a natural, unstimulated manner, and collected into sterile samples. Also in saliva, higher concentrations of MMP-8 are found in patients with periodontitis compared to healthy individuals [106,112]. MMP-8 concentrations in the saliva of patients with periodontitis were four times higher than in healthy individuals [97]. However, regardless of the form, periodontal patients had higher concentrations and activity levels of this enzyme than healthy subjects.

Similar to previous studies, MMP-8 concentrations in saliva correlate with indicators of periodontitis. MMP-8 has been shown to be positively correlated with plaque index [117], the presence of bleeding during probing, and greater probing depth [99,118]. Importantly, salivary MMP-8 concentrations were higher in patients with more advanced periodontitis, and Receiver Operating Characteristic (ROC) curve analysis showed that this enzyme has high diagnostic power in detecting periodontitis [118]. This was also confirmed by Mohammed et al. [119]. However, it should be noted that a single study indicates that the determination of MMP-8 concentration is not suitable for differentiating the stages of periodontitis [90]. In addition, in the saliva of people with periodontitis, MMP-8 is found in different forms, i.e., in the active form, bound in complexes, pro-MMP-8 of neutrophilic origin, and mesenchymal origin [120].

Periodontal treatments also affect MMP-8 concentrations in patients’ saliva. The use of non-invasive treatments such as scaling, curettage, and professional tooth brushing was associated with a decrease in MMP-8 concentrations in patients—a decrease that was seen at 3 weeks after the treatments [119]. With doxycycline treatment, however, there is no decrease in MMP-8 concentration in patients’ saliva [121].

Different from saliva is the fluid obtained from oral rinse. Oral rinse fluid was obtained from patients who rinsed their mouths with water for 30 min and then collected into a sterile vessel. As reported by Yuan et al. [103], MMP-8 concentrations in oral rinse fluid are not statistically different between patients with periodontitis and healthy individuals. 

On the other hand, however, an enormous amount of research indicates the considerable usefulness of the “Active-Matrix Metalloproteinase-8 Point-of-Care (PoC)/Chairside Mouthrinse Test” in the diagnosis of periodontal disease. This is the first commercially available test based on the determination of aMMP-8. Such tests are in the form of preparations such as PerioSafe^®^ and ImplantSafe^®^; these preparations were invented in Finland and further developed in Germany. The technology is based on monoclonal antibodies and is now available in many countries [122]. The usefulness of this test has been confirmed in studies by Alassiri et al. [122], Sorsa et al. [123], Sorsa et al. [124], Räisänen et al. [125], Heikkinen et al. [126], Räisänen et al. [127], Schmalz et al. [128], Raivisto et al. [129], Lähteenmäki et al. [130], and Öztürk et al. [131]. According to most studies, patients with periodontitis had higher levels of aMMP-8 compared to healthy subjects [124,131], and aMMP-8 determination showed higher sensitivity and specificity, and fewer false positives than traditional methods of diagnosing periodontitis (conventional bleeding on probing BOP), and was more resistant to the interfering effects of oral hygiene [124,127,131].

Importantly, aMMP-8 correlated with the severity of periodontitis [128,131], PI index [124], and also the presence of Porphyromonas gingivalis, Tannerella forsythia, Prevotella intermedia, Parvimonas micra, Camphylobacter rectus, and Eubacterium nodatum [128]. A single study also indicates the utility of aMMP-8 in assessing the efficacy of periodontal treatment; patients showed lower levels of aMMP-8 after treatment [129]. According to these studies, MMP-8 has great potential in detecting periodontitis compared to traditional methods, but it must be assayed with oral rinse fluid in its active form. A summary of the role of MMP-8 in different types of biological material is shown in Table 1.

## 6. MMP-8 Gene Polymorphisms and Periodontitis

Some work has also addressed the link between MMP-8 polymorphisms and the risk of periodontitis, but this topic is currently poorly studied and contradictory. The first MMP-8 polymorphism to be studied is MMP-8-799C/T. As reported by Weng et al. [37], Chou et al. [134], and Emingil et al. [135], patients with this polymorphism have an increased tendency to develop periodontitis (including generalized aggressive periodontitis), but on the other hand, a single study indicates that there is no relationship between the presence of the MMP-8-799C/T polymorphism and the risk of periodontitis [136]. Other polymorphisms that have been studied include MMP-8+17C/G. According to most studies, the presence of this polymorphism was not associated with the occurrence of periodontitis [136,137,138,139]. Again, however, a single study indicates that a higher frequency of MMP-8+17C/G is associated with a higher incidence of periodontitis [140]. One team also studied the MMP-8-381 A/G polymorphism; however, in this case, no association with periodontitis was found [137].

## 7. Conclusions

Nowadays, the number of people who are or will be affected by periodontitis in the future is steadily increasing. The onset of this disease is influenced by many factors, some of which are impossible to eliminate. It is estimated that the disease is the most common cause of tooth loss. Considering the increasing number of periodontal patients, new tools are being sought to diagnose and assess the severity of periodontitis and also to evaluate the effectiveness of its treatment and monitor the risk of recurrence. Such a modern tool may be MMP-8, an enzyme whose elevated concentrations are observed both in the gingival fluid and saliva of patients with periodontitis. The determination of MMP-8 concentrations can be useful not only in the detection of periodontal disease, but also in the evaluation of some clinical parameters of this condition and in assessing the effectiveness of periodontal treatments. However, unequivocally determining the usefulness of MMP-8 as an auxiliary tool in periodontology requires more research. 

Importantly, in the case of aMMP-8 assayed with Active-Matrix Metalloproteinase-8 Point-of-Care (PoC)/Chairside Mouthrinse Tests such as PerioSafe^®^ or ImplantSafe^®^, this enzyme shows high potential not only in diagnosing periodontitis but also in determining its severity and assessing the effectiveness of treatment. However, consideration should also be given to determining the usefulness of aMMP-8 in other types of biological material in periodontal patients.

## Figures and Tables

**Figure 1 ijms-25-02721-f001:**
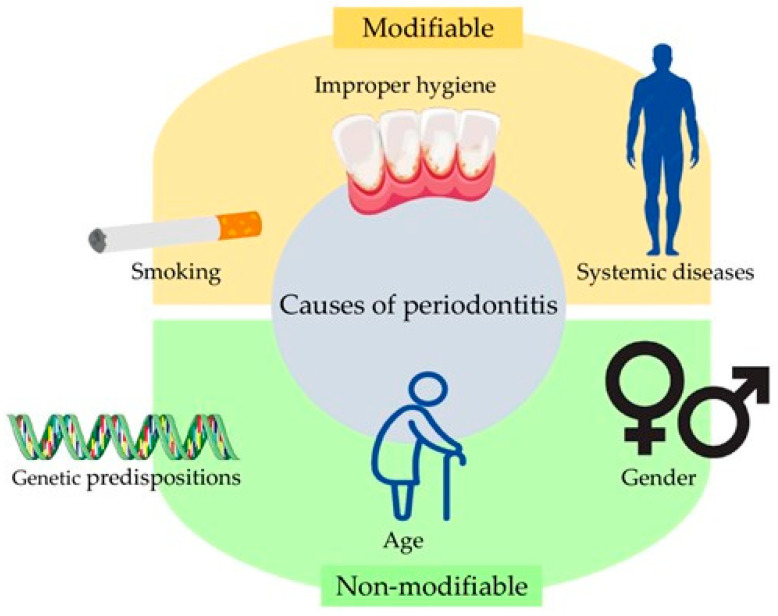
The most important causes of periodontitis.

**Figure 2 ijms-25-02721-f002:**
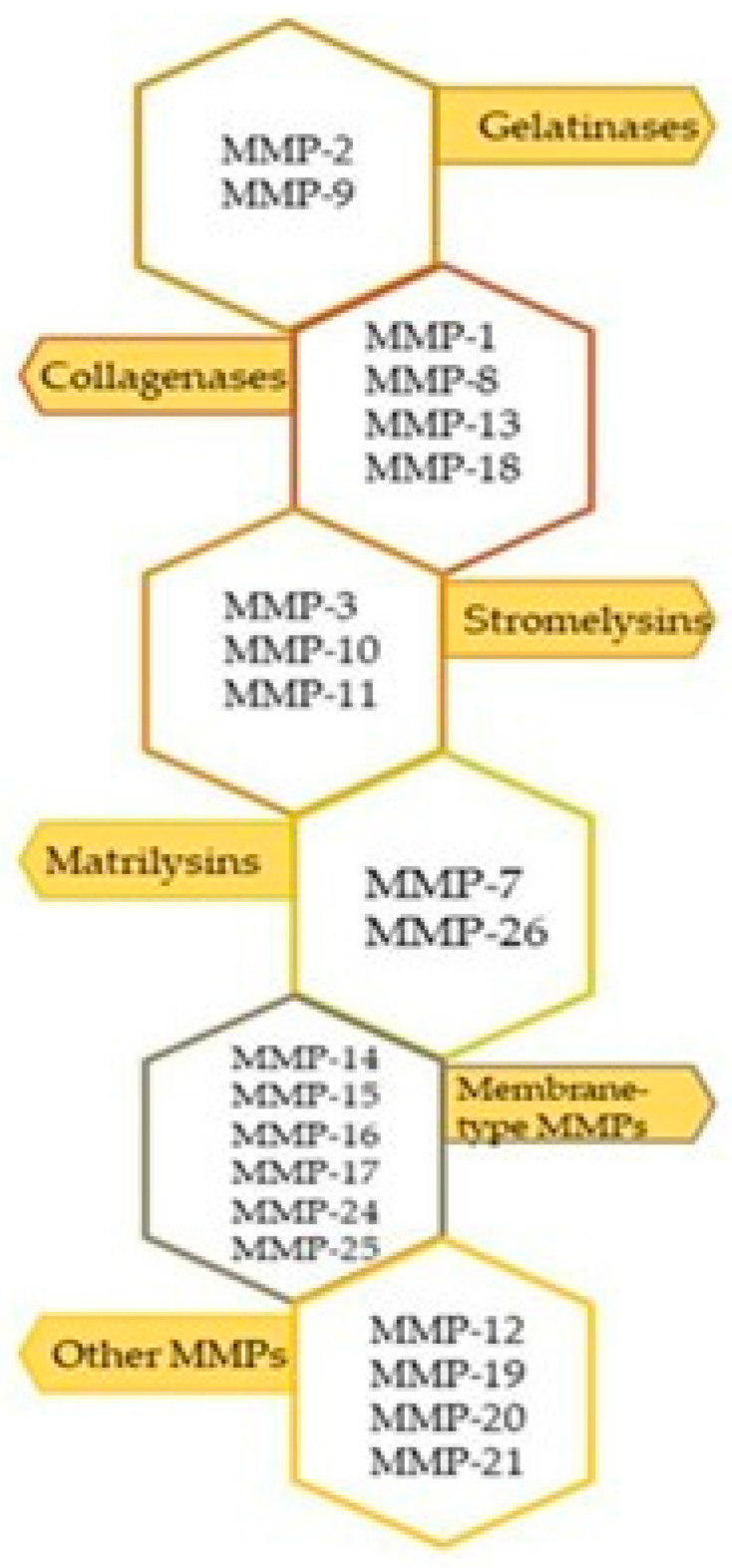
Division of MMPs by substrate spectrum.

**Figure 3 ijms-25-02721-f003:**
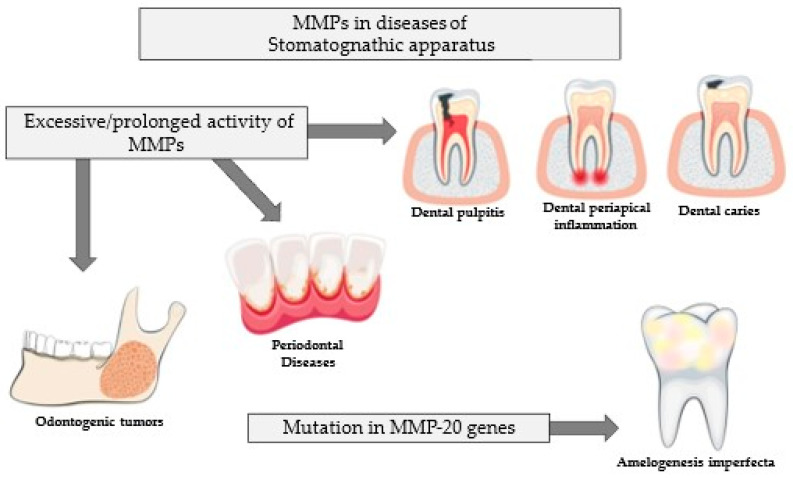
MMPs in stomatognathic diseases.

**Table 1 ijms-25-02721-t001:** Summary of the usefulness of MMP-8/aMMP-8 in different types of biological material in patients with periodontitis.

Summary of the Usefulness of MMP-8 in Different Types of Biological Material in Patients with Periodontitis
Gingival fluid
- ↑ concentrations of MMP-8 in periodontitis patients [93,94,96,97,98,99,100,101,102,103,104,105,106,107]. - Concentration of MMP-8 3.6 times ↑ in periodontitis patients than in healthy controls [97]. - ↑ concentrations of MMP-8 in patients with more severe forms of the disease [93,99,101]. **Contradictory reports on the possibility of differentiating types of periodontitis.** - Determination of MMP-8 concentrations not useful for differentiating types of periodontitis [100]. - Determination of MMP-8 is useful in differentiating serve and mild periodontitis [93]. - MMP-8 concentrations correlate with CAL, PI [97,108], increased depth of gingival pockets, and bleeding during probing [96,99,106,108]. - MMP-8 expression correlates strongly with CAL [97]. - In smokers, high levels of MMP-8 correlate with poorer response to periodontal treatment [109]. - Concentration of MMP-8 correlates with the presence of *Fusobacterium nucleatum* [101], *Tannerella forsythia*, and *Treponema denticola* [110]. - ↓ MMP-8 levels after periodontal treatment, antibiotics, and meloxicam use [106,112,114,116]. - ↓ MMP-8 levels after non-invasive hygiene methods [104,108,112,113,115].
Saliva
- ↑ concentrations of MMP-8 in periodontitis patients [97,99,107,116,117,118,119,132,133]. - ↑ concentrations of MMP-8 in patients with more severe forms of the disease [118]. - High diagnostic power of MMP-8 in detecting periodontitis [118,119]. - MMP-8 concentrations correlate with PI [117], increased depth of gingival pockets, and bleeding during probing [99,118]. - Determination of MMP-8 concentrations is not suitable for differentiating stage of periodontitis [90]. - ↓ MMP-8 levels after non-invasive hygiene methods [119]. - No change in MMP-8 levels after doxycycline therapy [121].
Oral rinse fluid
- No differences in MMP-8 concentrations between patients with periodontitis and healthy subjects [103]. - ↑ concentrations of aMMP-8 in patients with periodontitis [124,131]. - ↑ sensitivity and specificity of aMMP-8 in detecting periodontitis compared to classical diagnostic methods (conventional bleeding on probing [BOP]) [29,124,131]. - False positives in the detection with aMMP-8 of periodontitis compared to BOP [127]. - Correlation of aMMP-8 with the severity of periodontitis [128,131], PI [124], and the presence of Porphyromonas *gingivalis*, *Tannerella forsythia*, *Prevotella intermedia*, *Parvimonas micra*, *Camphylobacter rectus*, and *Eubacterium nodatum* [128]. - ↓ of aMMP-8 after periodontal treatment [129].

↑—increase; ↓—decrease.

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
