# Peer review of "Importance of Metalloproteinase 8 (MMP-8) in the Diagnosis of Periodontitis"

_ijms, 2024, doi:10.3390/ijms25052721_

Round 1

Reviewer 1 Report

Comments and Suggestions for Authors

This is an interesting paper, however, there are several points that could be addressed to make this paper more appropriate. First, the fact that it is a review that is not presented accordingly, with the recognition of the several biases that any author would have when facing an increasing number of papers as in the case of this subject, and the authors simply do not mention this problem and do not discuss how they decided to use some papers and not other published at the same time or even earlier.

But overall the paper is very well written and certainly will contribute to the field.

Some specific points. 

When references 13-15 are cited for the first time, the authors do not cite specific references after each condition. Having followed the literature of MMPs over the years (and made several searches over the decades), I would say that there are many more papers on MMPs and cardiovascular diseases nowadays than papers on MMPs and cancer. And by reading that part where the references 13-15 are cited, one has the impression that MMP inhibition is still a valid proposal for today in the battle against cancer. And this is not the case. There are hundreds to thousands of papers on MMPs and cardiovascular diseases, including sevaral reviews. 

My strong suggestion is to change this, since this review might be misleading. I think that this is actually happening because no concerns about a literature search was made, and this has to be acknowledged by the authors in the beginning or in the end of the paper.

The authors then describe the risk factors for having increased amounts of MMPs, the differences in gender (due to hormons) and genes. Good job. However, why have polymorphisms in the very genes that encode MMPs have not been cited.

Line 101.

These MMP polymorphisms are very important to confer a particular risk factor for many diseases, particularly cardiovascular and renal disease. Please include some references. Polymorphisms of MMP-2 are likely a good start for the search.

Ther graph for MMP classes is so interesting. Congratulations.

lines 176-180. 

References should be cited close to each specific point, and not after the whole long sentence.

Line 192-193: MMP-2 or MMP-8 ?

line 194: or MMP-13 (it is not clear to me what is meant)

Legend of Figure 3. Please make a proper description of the figure. Only the protein structure does not contribute any information to this paper. Please incluse the fact that the ions are observable, explain the main structure of MMP-8 as a collagenase and cite the source of this picture.

The description of the physiological substrates of MMP-8 is very good, in my opinion. However, a very important information appear to be missing. And that is the fact that collagenases are supposed to be such essential proteases for inflammation and cell movement because they are the solely or almost the solely proteases that can cleave the native fibrillar collagens triple-hellix, and only thereafter other proteases are more effective. Or has this changed? I think this is very important, particularly for processes that involve breakdown of large volumes of collagen-containing tissues, and this might be the case of morphogenesis, and it certainly is the case for the breakdown of gengiva and periodontal ligament, where very large/tuff/robust  bundles of collagen exist. 

line 217   please change the place of the "such". Correct is " many diseases such as..."

When references 85-90 are cited, to me there are many references from the group of Dr. Timo Sorsa missing, and well as the papers by Dr. Andrea Marcaccini, that showed increased MMP-8 in untreated periodontal patients, and that MMP-8 and also MMP-9 decreased after non-surgical periodontal treatment. Those authors published also correlations of these MMPs with mieloperoxidase and even calculated the number of PMN neutrophils that left the diseased pockets, based on MPO activity, much earlier than the cited paper (paper 112, published in 2014 or 2022 (I am not sure about my writing now), and it did not calculated the number of cells etc...).  So, this needs to be changed. I know a review is not an easy task, but the misscitation of papers, and the lack of citation of the papers where important information was shown for the first time is very sad to see, and it greatly compromises the value of the work of scientists, whose work relies in the discoveries of the whole community.

Reference 94: there is a mistake in the writing of the auther Hernandez (it seems to me at least)

line 312 please explain the active MMP-8 and then use the abbreviation aMMP-8. As it is, the reader cannot follow.

Nice job !

Author Response

Reviewer 1

This is an interesting paper, however, there are several points that could be addressed to make this paper more appropriate. First, the fact that it is a review that is not presented accordingly, with the recognition of the several biases that any author would have when facing an increasing number of papers as in the case of this subject, and the authors simply do not mention this problem and do not discuss how they decided to use some papers and not other published at the same time or even earlier.

But overall the paper is very well written and certainly will contribute to the field.

Dear Reviewer,

We would like to thank you very much for your careful review of our paper, entitled. " Importance of metalloproteinase 8 (MMP-8) in the diagnosis of periodontitis" and for your accurate and useful suggestions. We have highlighted responses to suggestions in blue italics while changes to the manuscript have been highlighted in green. We hope that the corrections made will prove satisfactory and allow publication of our work in the “International Journal of Molecular Sciences”. While preparing the paper, we performed a thorough review of the literature, using a number of internecine databases such as PubMed, Google Scholar and ResearchGate. We tried to select the latest scientific articles for our work. If we have omitted certain topics that the reviewer believes are relevant, we will try to add them after first indicating the missing subject matter.

When references 13-15 are cited for the first time, the authors do not cite specific references after each condition. Having followed the literature of MMPs over the years (and made several searches over the decades), I would say that there are many more papers on MMPs and cardiovascular diseases nowadays than papers on MMPs and cancer. And by reading that part where the references 13-15 are cited, one has the impression that MMP inhibition is still a valid proposal for today in the battle against cancer. And this is not the case. There are hundreds to thousands of papers on MMPs and cardiovascular diseases, including sevaral reviews.

My strong suggestion is to change this, since this review might be misleading. I think that this is actually happening because no concerns about a literature search was made, and this has to be acknowledged by the authors in the beginning or in the end of the paper.

First of all, thank you for your rightful comment. We have made the appropriate corrections so that the references cite a specific type of disease. In addition, we have changed the source literature so that the papers we cite refer only to the role of MMPs in a particular type of disease. The changes have been highlighted in green in the manuscript. However, we would like to point out that in the manuscript we do not write about the potential in blocking the activity of MMPs in the treatment of cancer, we only mention that these enzymes show potential as modern tumor markers which is confirmed by numerous studies conducted by our scientific team. However, we agree with the reviewer that blocking the activity of MMPs in cancer currently has low potential for treating this group of conditions.

The authors then describe the risk factors for having increased amounts of MMPs, the differences in gender (due to hormons) and genes. Good job. However, why have polymorphisms in the very genes that encode MMPs have not been cited.

Line 101.

These MMP polymorphisms are very important to confer a particular risk factor for many diseases, particularly cardiovascular and renal disease. Please include some references. Polymorphisms of MMP-2 are likely a good start for the search.

Thank you for your positive comment and good point about polymorphisms in genes encoding MMPs. This is an oversight on our part for which we apologize, we agree that, especially in relation to the topic of the paper, this issue should be discussed. We have modified the passage in question accordingly, enriching it with information on MMPs polymorphisms in periodontal disease. We have also modified the literature accordingly. In addition, thanks to a reviewer's suggestion, we have added a chapter devoted exclusively to the role of polymorphisms in genes encoding MMP-8 in the development of periodontitis titled "MMP-8 gene polymorphisms and periodontitis." All changes have been highlighted in green in the manuscript.

List of added literature:

  1. 134. Chou, Y.H.; Ho, Y.P.; Lin, Y.C.; Hu, K.F.; Yang, Y.H.; Ho, K.Y.; Wu, W.M.; Hsi, E.; Tsai, C.C. MMP-8 -799 C>T genetic polymorphism is associated with the susceptibility to chronic and aggressive periodontitis in Taiwanese. J Clin Periodontol 2011, 38, 1078-84.

  2. 135. Emingil, G.; Han, B.; Gürkan, A.; Berdeli, A.; Tervahartiala, T.; Salo, T.; Pussinen, P.J.; Köse, T.; Atilla, G.; Sorsa, T. Matrix Metalloproteinase (MMP)‐8 and Tissue Inhibitor of MMP‐1 (TIMP‐1) Gene Polymorphisms in Generalized Aggressive Periodontitis: Gingival Crevicular Fluid MMP‐8 and TIMP‐1 Levels and Outcome of Periodontal Therapy. Journal of Peri-odontology 2014, 85, 1070–1080, doi:10.1902/jop.2013.130365.

  3. 136. Izakovicova Holla, L.; Hrdlickova, B.; Vokurka, J.; Fassmann, A. Matrix Metalloproteinase 8 (MMP8) Gene Polymorphisms in Chronic Periodontitis. Archives of Oral Biology 2012, 57, 188–196, doi:10.1016/j.archoralbio.2011.08.018.

  4. 137. Li, W.; Zhu, Y.; Singh, P.; Ajmera, D.H.; Song, J.; Ji, P. Association of Common Variants in MMPs with Periodontitis Risk. Disease Markers 2016, 2016, 1–20, doi:10.1155/2016/1545974.

  5. 138. Heikkinen, A.M.; Kettunen, K.; Kovanen, L.; Haukka, J.; Elg, J.; Husu, H.; Tervahartiala, T.; Pussinen, P.; Meurman, J.; Sorsa, T. Inflammatory Mediator Polymorphisms Associate with Initial Periodontitis in Adolescents. Clinical & Exp Dental Res 2016, 2, 208–215, doi:10.1002/cre2.40.

  6. 139. Putri, H.; Sulijaya, B.; Hartomo, B.T.; Suhartono, A.W.; Auerkari, E.I. +17 C/G Polymorphism in Matrix Metalloproteinase (MMP)-8 Gene and Its Association with Periodontitis. jos 2020, 73, 154–158, doi:10.5114/jos.2020.98310.

  7. 140. Majumder, P.; Ghosh, S.; Dey, S.K. Matrix metalloproteinase gene polymorphisms in chronic periodontitis: a case-control study in the Indian population. J Genet 2019, 98, 32.

Ther graph for MMP classes is so interesting. Congratulations.

Thank you very much for your positive opinion.

lines 176-180.

References should be cited close to each specific point, and not after the whole long sentence.

Thank you for your comment. References have been corrected as suggested by the reviewer. The relevant changes are marked in green.

Line 192-193: MMP-2 or MMP-8?

Thank you for your kind comment and we apologize for the mistake. The error has been corrected and the change highlighted in green.

line 194: or MMP-13 (it is not clear to me what is meant)

The given fragment explains that the gene encoding MMP-8 is located in the same cluster as other genes encoding MMPs such as MMP-1 and MMP-13. A cluster is a group of closely related genes that encode closely related proteins.

Legend of Figure 3. Please make a proper description of the figure. Only the protein structure does not contribute any information to this paper. Please incluse the fact that the ions are observable, explain the main structure of MMP-8 as a collagenase and cite the source of this picture.

Thank you for the right comment. The reviewer rightly pointed out that the figure showing the spatial structure of MMP-8 is described incorrectly and does not contribute relevant information to the article. After consulting with all the authors of this article, we have decided to remove figure number 3.

The description of the physiological substrates of MMP-8 is very good, in my opinion. However, a very important information appear to be missing. And that is the fact that collagenases are supposed to be such essential proteases for inflammation and cell movement because they are the solely or almost the solely proteases that can cleave the native fibrillar collagens triple-hellix, and only thereafter other proteases are more effective. Or has this changed? I think this is very important, particularly for processes that involve breakdown of large volumes of collagen-containing tissues, and this might be the case of morphogenesis, and it certainly is the case for the breakdown of gengiva and periodontal ligament, where very large/tuff/robust  bundles of collagen exist. 

Thank you for your comment. Since none of the co-authors study protein biology at such an advanced level, we did not include the above mechanism in our paper. However, we did review the scientific literature on the degradation of native collagen by MMPs. As suggested by the reviewer, some enzymes in this group are capable of native fibrillar collagens triple-hellix, however, the bulk of the descriptions mainly point to MMP-1 as the enzyme responsible for this degradation [Literature: Sprangers S, Everts V. Molecular pathways of cell-mediated degradation of fibrillar collagen. Matrix Biol. 2019 Jan;75-76:190-200. doi: 10.1016/j.matbio.2017.11.008. epub 2017 Nov 21. PMID: 29162487; Ala-aho R, Kähäri VM. Collagenases in cancer. Biochimie. 2005 Mar-Apr;87(3-4):273-86. doi: 10.1016/j.biochi.2004.12.009. PMID: 15781314.]. However, since studies also indicate that MMP-8 is involved in the above-described process, we have supplemented the section on the physiological action of this enzyme with the information provided, as suggested by the reviewer. The changes have been highlighted in green.

line 217   please change the place of the "such". Correct is " many diseases such as..."

Thank you for your comment and we apologize for the mistake. We have made changes to the manuscript. The changes have been highlighted in green.

When references 85-90 are cited, to me there are many references from the group of Dr. Timo Sorsa missing, and well as the papers by Dr. Andrea Marcaccini, that showed increased MMP-8 in untreated periodontal patients, and that MMP-8 and also MMP-9 decreased after non-surgical periodontal treatment. Those authors published also correlations of these MMPs with mieloperoxidase and even calculated the number of PMN neutrophils that left the diseased pockets, based on MPO activity, much earlier than the cited paper (paper 112, published in 2014 or 2022 (I am not sure about my writing now), and it did not calculated the number of cells etc...).  So, this needs to be changed. I know a review is not an easy task, but the misscitation of papers, and the lack of citation of the papers where important information was shown for the first time is very sad to see, and it greatly compromises the value of the work of scientists, whose work relies in the discoveries of the whole community.

Thank you for your attention and we apologize for the shortcomings on our part. We regret to admit that several relevant articles were not included in our paper. Thanks to the reviewer's attention, we have again reviewed the literature related to MMP-8 in periodontitis and included several additional articles in the paper. Once again, we would like to apologize for such serious oversights on our part. In the case of the work of the Marcaccini et al. team, we are aware that they also studied serum concentrations of MMP-8 in patients with periodontitis; however, we chose not to add this work to the manuscript. This is mainly due to the type of material that is serum because in the paper we limited ourselves to biological material collected only from the oral cavity (gingival fluid, saliva, etc.). In addition, the need for medical personnel other than dentists to collect blood greatly limits the usefulness of this material as a tool in dentistry.

List of added articles:

  1. Marcaccini, A.M.; Meschiari, C.A.; Zuardi, L.R.; De Sousa, T.S.; Taba, M.; Teofilo, J.M.; Jacob‐Ferreira, A.L.B.; Tanus‐Santos, J.E.; Novaes, A.B.; Gerlach, R.F. Gingival Crevicular Fluid Levels of MMP‐8, MMP‐9, TIMP‐2, and MPO Decrease after Periodontal Therapy. J Clinic Periodontology 2010, 37, 180–190, doi:10.1111/j.1600-051X.2009.01512.x.

  2. Chen, H.Y.; Cox, S.W.; Eley, B.M.; Mäntylä, P.; Rönkä, H.; Sorsa, T. Matrix Metalloproteinase‐8 Levels and Elastase Activities in Gingival Crevicular Fluid from Chronic Adult Periodontitis Patients. J Clinic Periodontology 2000, 27, 366–369, doi:10.1034/j.1600-051x.2000.027005366.x.

  3. Ingman, T.; Tervahartiala, T.; Ding, Y.; Tschesche, H.; Haerian, A.; Kinane, D.F.; Konttinen, Y.T.; Sorsa, T. Matrix Metalloproteinases and Their Inhibitors in Gingival Crevicular Fluid and Saliva of Periodontitis Patients. J Clinic Periodontology 1996, 23, 1127–1132, doi:10.1111/j.1600-051X.1996.tb01814.x.

  4. Sorsa, T.; Alassiri, S.; Grigoriadis, A.; Räisänen, I.T.; Pärnänen, P.; Nwhator, S.O.; Gieselmann, D.-R.; Sakellari, D. Active MMP-8 (aMMP-8) as a Grading and Staging Biomarker in the Periodontitis Classification. Diagnostics 2020, 10, 61, doi:10.3390/diagnostics10020061.

  5. Buduneli, N.; Vardar, S.; Atilla, G.; Sorsa, T.; Luoto, H.; Baylas, H. Gingival Crevicular Fluid Matrix Metalloproteinase‐8 Levels Following Adjunctive Use of Meloxicam and Initial Phase of Periodontal Therapy. Journal of Periodontology 2002, 73, 103–109, doi:10.1902/jop.2002.73.1.103.

Reference 94: there is a mistake in the writing of the auther Hernandez (it seems to me at least)

Thank you for your attention. The surname has been corrected.

Line 312 please explain the active MMP-8 and then use the abbreviation aMMP-8. As it is, the reader cannot follow.

Thank you for your comment and we apologize for the mistake. We have made changes to the appropriate use of the abbreviation of aMMP-8. All changes have been highlighted in green.

In addition, thanks to the reviewer's suggestions, we have made changes regarding the role of Active-Matrix Metalloproteinase-8 Point-of-Care (PoC)/Chairside Mouthrinse Test. This topic was brought to us by the reviewer's suggestion regarding the work of Dr. T.Sorsa who is pioneering the introduction of such assays. Immunoassays that allow the determination of aMMP-8 in oral rinse fluid were invented in Finland, and are present in many countries. The most commonly used kits are PerioSafe® or ImplantSafe®.

Accordingly, the manuscript was modified accordingly and the literature was enriched with the following papers:

  1. Alassiri, S.; Parnanen, P.; Rathnayake, N.; Johannsen, G.; Heikkinen, A.-M.; Lazzara, R.; Van Der Schoor, P.; Van Der Schoor, J.G.; Tervahartiala, T.; Gieselmann, D.; et al. The Ability of Quantitative, Specific, and Sensitive Point-of-Care/Chair-Side Oral Fluid Immunotests for aMMP-8 to Detect Periodontal and Peri-Implant Diseases. Disease Markers 2018, 2018, 1–5, doi:10.1155/2018/1306396.

  2. Sorsa, T.; Gursoy, U.K.; Nwhator, S.; Hernandez, M.; Tervahartiala, T.; Leppilahti, J.; Gursoy, M.; Könönen, E.; Emingil, G.; Pussinen, P.J, Analysis of Matrix Metalloproteinases, Especially MMP‐8, in Gingival Crevicular Fluid, Mouthrinse and Saliva for Monitoring Periodontal Diseases. Periodontology 2000 2016, 70, 142–163, doi:10.1111/prd.12101.

  3. Sorsa, T.; Alassiri, S.; Grigoriadis, A.; Räisänen, I.T.; Pärnänen, P.; Nwhator, S.O.; Gieselmann, D.-R.; Sakellari, D. Active MMP-8 (aMMP-8) as a Grading and Staging Biomarker in the Periodontitis Classification. Diagnostics 2020, 10, 61, doi:10.3390/diagnostics10020061.

  4. Räisänen, I.; Sorsa, T.; Van Der Schoor, G.-J.; Tervahartiala, T.; Van Der Schoor, P.; Gieselmann, D.-R.; Heikkinen, A. Active Matrix Metalloproteinase-8 Point-of-Care (PoC)/Chairside Mouthrinse Test vs. Bleeding on Probing in Diagnosing Subclinical Periodontitis in Adolescents. Diagnostics 2019, 9, 34, doi:10.3390/diagnostics9010034.

  5. Heikkinen, A.M.; Raivisto, T.; Kettunen, K.; Kovanen, L.; Haukka, J.; Pakbaznejad Esmaeili, E.; Elg, J.; Gieselmann, D.; Rathnayake, N.; Ruokonen, H.; et al. Pilot Study on the Genetic Background of an Active Matrix Metalloproteinase‐8 Test in Finnish Adolescents. Journal of Periodontology 2017, 88, 464–472, doi:10.1902/jop.2016.160441.

  6. Räisänen, I.; Heikkinen, A.; Siren, E.; Tervahartiala, T.; Gieselmann, D.-R.; Van Der Schoor, G.-J.; Van Der Schoor, P.; Sorsa, T. Point-of-Care/Chairside aMMP-8 Analytics of Periodontal Diseases’ Activity and Episodic Progression. Diagnostics 2018, 8, 74, doi:10.3390/diagnostics8040074.

  7. Schmalz, G.; Hübscher, A.E.; Angermann, H.; Schmidt, J.; Schmickler, J.; Legler, T.J.; Ziebolz, D. Associations of Chairside Salivary aMMP-8 Findings with Periodontal Parameters, Potentially Periodontal Pathogenic Bacteria and Selected Blood Parameters in Systemically Healthy Adults. Diagnostic Microbiology and Infectious Disease 2019, 95, 179–184, doi:10.1016/j.diagmicrobio.2019.05.006.

  8. Raivisto, T.; Sorsa, T.; Räisänen, I.; Kauppila, T.; Ruokonen, H.; Tervahartiala, T.; Haukka, J.; Heikkinen, A.M. Active Matrix Metalloproteinase-8 Chair Side Mouth Rinse Test, Health Behaviour and Oral Health in Finnish Adolescent Cohort. J. Clin. Diagn. Res. 2019

  9. Lähteenmäki, H.; Pätilä, T.; Pärnänen, P.; Räisänen, I.; Tervahartiala, T.; Gupta, S.; Sorsa, T. aMMP-8 Point-of-Care - Diagnostic Methods and Treatment Modalities in Periodontitis and Peri-Implantitis. Expert Opinion on Therapeutic Targets 2023, 27, 627–637, doi:10.1080/14728222.2023.2240014.

  10. Öztürk, V.Ö.; Emingil, G.; Umeizudike, K.; Tervahartiala, T.; Gieselmann, D.-R.; Maier, K.; Köse, T.; Sorsa, T.; Alassiri, S. Evaluation of Active Matrix Metalloproteinase-8 (aMMP-8) Chair-Side Test as a Diagnostic Biomarker in the Staging of Periodontal Diseases. Archives of Oral Biology 2021, 124, 104955, doi:10.1016/j.archoralbio.2020.104955.

Again, we thank the reviewer for all the guidance and corrections. It is our hope that the manuscript, after revision, will meet the reviewer's expectations and be published in the International Journal of Molecular Sciences.

Best regards,

prof. dr hab. Sławomir Ławicki

also on behalf of all authors

Reviewer 2 Report

Comments and Suggestions for Authors

The work by Zalewska et al. concerns the importance of Metalloproteinase 8 (MMP-8) in the diagnosis of periodontitis. The topic of this manuscript is very important and current. The work is thoughtfully organized and composed, drawing on recent research for support. But before the updated document can be processed further, a few changes need to be made to it:
1. it is worth enriching the introduction with a graphic study of the causes of the disease

2. please indicate the innovativeness of the work and how this work differs from other already published ones (e.g. 10.3390/ijms23031806, 10.1097/MD.0000000000009642)

3. Please specify what is the role of MMPs in stomatognathic diseases shown in Figure 2

4. please describe how you can obtain MMP-8 in gingival fluid, saliva and oral rinse fluid. What should be the clinical utility protocol? Can such tests be performed in a dentist's office?

5. Table 1 should be placed in chapter 5 rather than 6

6. please rewrite the conclusions after taking into account the above-mentioned comments and presenting the clinical utility of the marker

Author Response

Reviewer 2

The work by Zalewska et al. concerns the importance of Metalloproteinase 8 (MMP-8) in the diagnosis of periodontitis. The topic of this manuscript is very important and current. The work is thoughtfully organized and composed, drawing on recent research for support. But before the updated document can be processed further, a few changes need to be made to it:

Dear Reviewer,

We would like to thank you very much for your careful review of our paper, entitled. " Importance of metalloproteinase 8 (MMP-8) in the diagnosis of periodontitis" and for your accurate and useful suggestions. We have highlighted responses to suggestions in blue italics while changes to the manuscript have been highlighted in green. We hope that the corrections made will prove satisfactory and allow publication of our work in the “International Journal of Molecular Sciences”.

1. it is worth enriching the introduction with a graphic study of the causes of the disease.

Thank you for the right comment. As suggested by the reviewer, we have added a figure to the manuscript showing the causes of periodontitis. The figure has been given the number 1 and is titled "The most important causes of periodontitis”. Due to the preparation of the new figure, the remaining figures will be renumbered as indicated in green.

2. please indicate the innovativeness of the work and how this work differs from other already published ones (e.g. 10.3390/ijms23031806, 10.1097/MD.0000000000009642)

Thank you for your comment. At the same time, we would like to point out that we understand the reviewer's concerns. Our work partially addresses the topics discussed earlier; however, several important points make it innovative.

First, in contrast to the work of Luchian et al. (10.3390/ijms23031806) despite focusing only on one selected metalloproteinase in our review, we presented many more source reports on the role of this enzyme in periodontitis. In addition, a huge plus of our work is the presentation of the collected results in a summary table which, in our opinion, is convenient for the potential reader and is a good complement to the text.

Second, unlike the work of Zhang et al. (10.1097/MD.0000000000009642) our paper is an opinion piece, not a meta-analysis.

Third, unlike other works of this type, in our manuscript we do not go straight to a description of the role of MMP-8 in periodontitis, but begin the paper with a description of the subject of the condition itself, thus making the paper a good source of information on the subject, so that it could be used by scientists and medics who are interested in the issue of periodontal disease.

Fourth, a huge plus of this work is the discussion of the role of MMPs in diseases of the stomatognathic apparatus and their expression in different types of oral cells. Currently, there are very few works summing up the knowledge on this subject, making our work innovative in this regard.

Fifth and finally, our work provides a very thorough description of MMP-8 physiology and the role of MMP-8 gene polymorphisms in periodontitis, which is a definite advantage of our manuscript.

3. Please specify what is the role of MMPs in stomatognathic diseases shown in Figure 2

Thank you for your attention. A more detailed description of the role of MMPs in the aforementioned diseases is included in the text. This figure depicts the diseases with which excessive/prolonged activity of MMPs is associated, while in the case of amelogenesis imperfecta, patients have mutations in the gene encoding MMP-20. For a better understanding of the figure, we have modified it accordingly, indicating a more precise role of MMPs in these diseases.

4. please describe how you can obtain MMP-8 in gingival fluid, saliva and oral rinse fluid.

First, thank you for your rightful comment. Adding a way to retrieve material is a great idea and will certainly enrich our work. We have modified portions of the manuscript accordingly to include this information. Any changes have been highlighted in green.

What should be the clinical utility protocol?

Again, thank you for your comment. The protocols for performance and material collection are established in advance by the reagent manufacturer and the dentist performing the procedure. With all due respect to the reviewer, we have chosen not to include descriptions of the protocols in the text of the manuscript because we believe that this would not enrich the content of the paper. The potential reader, reaching for our paper, will always be able to use the links that will direct him or her to the relevant source literature that contains a detailed description of the procedure performed.

Can such tests be performed in a dentist's office?

Due to the type of testing methodology used for MMP-8 assays, most tests cannot be determined in the dental office. Performing an immunosorbent assay (ELISA) or immunofluorometric assay (IFMA), with IFMA requiring professional equipment that is not on the equipment of most dental offices. In the case of aMMP-8, such a test can be performed in a dental office. The test is based on immunological methods and was invented in Finland, and is available in many countries. It is available in formulations such as PerioSafe® or ImplantSafe®. Such a test can be performed in any dental office and has very high sensitivity and specificity in detecting periodontitis. Our work has been enriched with a description of this methodology.

5. Table 1 should be placed in chapter 5 rather than 6

Thank you for your attention. The table has been moved as suggested by the reviewer.

6. please rewrite the conclusions after taking into account the above-mentioned comments and presenting the clinical utility of the marker

Thank you for your comment above. The conclusions have been reworded. In our opinion, the highest clinical utility is characterized by the determination of aMMP-8. In accordance with the reviewer's suggestions, we have developed the subject matter of this parameter and changed the conclusions accordingly. According to the conclusions, we have also made changes to the abstract.The changes have been highlighted in green in the text.

Again, we thank the reviewer for all the guidance and corrections. It is our hope that the manuscript, after revision, will meet the reviewer's expectations and be published in the International Journal of Molecular Sciences.

Best regards,

prof. dr hab. Sławomir Ławicki

also on behalf of all authors

Round 2

Reviewer 1 Report

Comments and Suggestions for Authors

Yes, the authors included references from the group of Dr. Timo Sorsa. However, please include the 3 references below.

https://doi.org/10.1016/j.cca.2013.03.008

doi: 10.1111/j.1600-051X.2009.01512.x.

  • DOI: 10.1016/j.cca.2009.09.012

As I mentioned in the other review, the authors cited a reference from 2017 describing the systemic increase of MMP-8. This was clearly shown in this paper a decade earlier. Furthermore, this paper has a fluorimetric assay done in plasma in which the total gelatinolytic is measured, and it decreases after therapy. Importantly, this is particularly relevant considering the potential of increased proteolytic activity to destabilize atherosclerotic plaques. No wonder from the 3 references cited, the one describing the MMPs in fluid and in plasma are highly cited so far (more than 200 citations). So, I do not see why they have not been included on a review on MMP-8 in periodontal disease. Except the fact that this review is not a systematic review, and so, picking up one paper and not another is unfortunately not a very scientific choice.

Please describe the fact that plasma concentrations of MMP-8 and MMP-9 and TIMP-2 increase in periodontal disease, and decrease after sclating and root planing, what does not happen with MMP-2 and TIMP-2. MMP-8 and MMP-9, as the author describe, are found in the granules of PMN neutrophils... as is MPO, a marker that Marcaccini at al also determined. marcaccini et al also have papers on MMP-8 and MPO in saliva and gingival fluid that should be cited, in my view, since the authors have cited references in this topic that were published later, and that were not so sound and did not cite the earlier work either. Sad to see.

Regards. 

Reviewer 2 Report

Comments and Suggestions for Authors

Accept in present form